# Hex1, the Major Component of Woronin Bodies, Is Required for Normal Development, Pathogenicity, and Stress Response in the Plant Pathogenic Fungus *Verticillium dahliae*

**DOI:** 10.3390/jof6040344

**Published:** 2020-12-07

**Authors:** Vasileios Vangalis, Ioannis A. Papaioannou, Emmanouil A. Markakis, Michael Knop, Milton A. Typas

**Affiliations:** 1Department of Genetics & Biotechnology, Faculty of Biology, National and Kapodistrian University of Athens, 15784 Athens, Greece; vasvagg@biol.uoa.gr; 2Center for Molecular Biology, Heidelberg University (ZMBH), 69120 Heidelberg, Germany; i.papaioannou@zmbh.uni-heidelberg.de (I.A.P.); m.knop@zmbh.uni-heidelberg.de (M.K.); 3Laboratory of Mycology, Department of Viticulture, Vegetable Crops, Floriculture and Plant Protection, Institute of Olive Tree, Subtropical Crops and Viticulture, N.A.G.R.E.F., Hellenic Agricultural Organization—DEMETER, 71307 Heraklion, Crete, Greece; markakis@nagref-her.gr; 4German Cancer Research Center (DKFZ), DKFZ-ZMBH Alliance, 69120 Heidelberg, Germany

**Keywords:** septal pore, hyphal integrity, virulence, ROS, heterokaryon incompatibility

## Abstract

Woronin bodies are membrane-bound organelles of filamentous ascomycetes that mediate hyphal compartmentalization by plugging septal pores upon hyphal damage. Their major component is the peroxisomal protein Hex1, which has also been implicated in additional cellular processes in fungi. Here, we analyzed the Hex1 homolog of *Verticillium dahliae*, an important asexual plant pathogen, and we report its pleiotropic involvement in fungal growth, physiology, stress response, and pathogenicity. Alternative splicing of the *Vdhex1* gene can lead to the production of two Hex1 isoforms, which are structurally similar to their *Neurospora crassa* homolog. We show that *Vd*Hex1 is targeted to the septum, consistently with its demonstrated function in sealing hyphal compartments to prevent excessive cytoplasmic bleeding upon injury. Furthermore, our investigation provides direct evidence for significant contributions of Hex1 in growth and morphogenesis, as well as in asexual reproduction capacity. We discovered that Hex1 is required both for normal responses to osmotic stress and factors that affect the cell wall and plasma-membrane integrity, and for normal resistance to oxidative stress and reactive oxygen species (ROS) homeostasis. The *Vdhex1* mutant exhibited diminished ability to colonize and cause disease on eggplant. Overall, we show that Hex1 has fundamentally important multifaceted roles in the biology of *V. dahliae*.

## 1. Introduction

The undifferentiated body (or thallus) of a typical filamentous fungus is the mycelium, a complex network of branched tubular cells called hyphae. In most fungi, these are partitioned into distinct cellular compartments by internal cross-walls, the septae [1,2]. Hyphal compartments are continuous with one another due to the occurrence of septal pores, which allow the intercellular flow of cytoplasm, including organelles, ensuring the rapid translocation of nutrients during colony establishment and facilitating the maintenance of cellular homeostasis during mycelial growth [3]. However, this ability of the mycelium to function as an integrated syncytium-like organism exposes it to a number of risks, e.g., excessive cytoplasmic leakage upon hyphal wounding or uncontrolled spread of selfish genetic elements (such as mycoviruses, transposable elements, and senescence plasmids) following vegetative hyphal fusion. Therefore, the intercompartmental traffic in hyphae must be highly regulated in order to protect the mycelium from such hazards, as well as to permit cellular heterogeneity and differentiation during developmental processes [3,4,5,6,7].

Early studies in various ascomycetes implicated a specialized membrane-bound vesicle described as Woronin body (WB) in sealing of septal pores to prevent cytoplasmic loss in response to hyphal damage [8,9]. It was later shown that WBs are involved in the control of intercellular communication and the maintenance of hyphal heterogeneity [10]. Woronin bodies, which are exclusively found in Pezizomycotina [9], originate from peroxisomes, and their biogenesis requires multiple peroxins [11]. They appear in electron microscopy images as electron-dense vesicles that are usually localized in close proximity to the septae, and they are rapidly tethered to the septal pore by interacting proteins following hyphal wounding [12,13]. At least 17 septal pore-associated WB-related proteins have been detected by mass spectrometry in *Neurospora crassa* [14].

The protein Hex1 was first characterized in *N. crassa* as the major component of WBs [15]. This protein spontaneously self-assembles into hexagonal crystals that comprise the dense core of WBs [16]. Conserved homologs of the *hex1* gene have been identified and studied in several members of the Pezizomycotina [17,18,19,20,21,22,23,24,25,26]. The gene presumably resulted from duplication of the ancestral gene encoding the eukaryotic initiation factor 5A (eIF-5A) [16]. After this duplication, *hex1* followed a different evolutionary trajectory that led to a new function by acquiring those amino acids that are responsible for its peroxisomal targeting and self-assembly. *Hex1* genes retain a conserved intron close to their N-terminus, and alternative splicing of this intron has been shown to produce two isoforms of the protein [17,18,19].

Deletion of *hex1* in several fungi generally led to excessive hyphal bleeding after wounding [20,27], as well as had pleiotropic effects on additional phenotypes associated with vegetative growth, asexual reproduction, and stress response against osmotic and cell wall-perturbating agents [24,25,26]. Regarding pathogenicity and virulence, conflicting results have been gathered from studies on plant, insect, and human pathogens. These range from important roles of Hex1 in the formation of appressoria (infection structures) and pathogenicity in *Magnaporthe grisea* [18] to moderately reduced or delayed virulence in *Fusarium graminearum* [22], *Aspergillus flavus* [25], and *A. fumigates* [13], and no significant defects in pathogenicity of *Colletotrichum orbiculare* [23] and *Metarhizium robertsii* [26].

The ascomycete *Verticillium dahliae* causes a wilt disease on a wide range of commercially important plants and crops, and it is responsible for enormous annual economic losses worldwide [28]. Its resting structures (microsclerotia), which can remain viable in the soil for several years, germinate upon induction by exudates from adjacent plant roots to form hyphae. This is followed by root penetration and colonization, leading the fungus to the host vascular system, through which it can cause systemic infection of the plant [29]. Except for the apical cells of its mycelium, hyphal compartments of *V. dahliae* are regularly septated, and, upon hyphal injury, WBs rapidly plug the pores of the flanking septae to seal damaged compartments [30,31]. This fungus is thought to completely lack a sexual stage and, thus, propagates exclusively through the dispersion of its asexual spores (conidia). A parasexual cycle, initiated by hyphal fusion between individuals with different genotypes to the formation of heterokaryons, has been described in *Verticillium* and is thought to increase genetic diversity through random chromosome assortment and frequent mitotic recombination [28,32]. Nevertheless, heterokaryon formation is restricted by incompatibility barriers [28,31], which seal and destroy incompatible fused cells, while the remaining mycelium remains unaffected [7,33].

This study aimed at the identification and functional characterization of *hex1* in fundamental biological processes of the important plant pathogen *V. dahliae*. For this, we deleted its *hex1* homolog to address the involvement of the gene in hyphal integrity, growth and development, pathogenicity, response to osmotic, cell wall-perturbating, and oxidative stress, and heterokaryon incompatibility. Our investigations revealed important roles of Hex1 in the biology of *V. dahliae*.

## 2. Materials and Methods

### 2.1. Fungal Strains, Culture Conditions, and Fungal DNA Isolation and Manipulation

All fungal strains that were constructed and used in this study are listed in Appendix A. Preparation and maintenance of monoconidial strains, culture media, and conditions were previously described [34].

Total genomic DNA of fungal strains was extracted according to published methods [34]. Standard and previously described procedures were followed for PCR amplification (all oligos are listed in Appendix A), cloning and maintenance of plasmids in *E. coli* strain DH5α (a list of plasmids is provided in Appendix A), restriction digestion, and Sanger sequencing [34,35].

### 2.2. Agrobacterium tumefaciens-Mediated Transformation of V. dahliae

The hypervirulent *A. tumefaciens* strain AGL-1 was transformed with binary plasmid vectors using a CaCl_2_/heat shock-mediated freeze–thaw method [36]. The resulting *A. tumefaciens* strains were used for transforming *V. dahliae* with protocols modified from [37]. After growth of *A. tumefaciens* in minimal medium (MM) supplemented with the proper selective antibiotics for 48 h (28 °C, 180 rpm), the culture was diluted to OD_600_ = 0.1 and incubated for 6 h in acetosyringone-containing induction medium (IM; 25 °C, 180 rpm, without antibiotics). Aliquots of this culture (100 μL) were then mixed with 100 μL aliquots of a conidial suspension of the desired *V. dahliae* strain (10^7^ conidia/mL, prepared from 7 day old cultures) and plated on sterile cellophane sheets on IM agar. The mixture was incubated for 48 h at 25 °C before its transfer to selective medium (potato dextrose agar, PDA) amended with 200 μg/mL cefotaxime, 50 μg/mL 5-fluorouracil, and 100 μg/mL geneticin and/or 15 μg/mL hygromycin B).

### 2.3. Protoplast Transformation of V. dahliae

At least 10^8^ conidia of the desired *V. dahliae* strain were incubated in Czapek-Dox complete medium (CzD-CM) for 16 h (25 °C, 175 rpm). Conidia/germlings were then pelleted, resuspended into a phosphate-KCl buffer (0.6 M KCl, 1/15 M KH_2_PO_4_, and 1/15 M Na_2_HPO_4_, pH 5.8; supplemented with 25 mM dithiothreitol (DTT) and incubated for 2 h at 30 °C (50 rpm). Protoplastation was performed by transferring the treated germlings to fresh phosphate-KCl buffer supplemented with 8 mg/mL of a lysing enzyme mix from *Trichoderma harzianum* (Sigma-Aldrich, St. Louis, MO, USA) and further incubating the mix at 30 °C (50 rpm) for 1–3 h. Protoplasts were separated from aggregated cellular debris by filtering through four layers of sterile gauge and resuspended into STC solution (1.2 M sorbitol, 10 mM Tris-Cl pH 7.5, and 50 mM CaCl_2_) at a final concentration of 10^8^/mL. Aliquots (100 μL) of this protoplast suspension were mixed with 2–8 μg of DNA and incubated for 30 min on ice before the addition of 200 μL of PTC buffer (40% PEG-6000, 25 mM Tris-Cl pH 7.5 and 25 mM CaCl_2_) and further incubation for 20 min on ice. This was followed by the stepwise addition of 1 mL of fresh PTC buffer and incubation for 15 min on ice. The mixture was then washed three times with STC buffer, resuspended into fresh STC buffer, and plated on regeneration agar medium (20.54 g sucrose, 0.1 g yeast extract, and 0.14 g casamino acids per 100 mL), overlaid with a sterile cellophane sheet. Following incubation at 25 °C for 24 h, the cellophane sheet was transferred onto the appropriate selective medium (PDA amended with 100 μg/mL geneticin and/or 15 μg/mL hygromycin B).

### 2.4. Deletion, Complementation, and sGFP-Tagging of V. dahliae hex1

The NEBuilder HiFi DNA Assembly Master Mix (New England Biolabs, Ipswich, MA, USA) was used (according to the manufacturer’s recommendations) for the construction, in two steps, of a plasmid vector for *hex1* deletion by homologous recombination. The first reaction resulted in a plasmid containing 2000 bp long homologous arms (amplified from the genomic DNA of *V. dahliae* isolate Ls.17 using the Herculase II Fusion DNA polymerase; Agilent, Santa Clara, CA, USA) flanking the *neo* cassette (conferring resistance to geneticin) from plasmid pSD1, in the backbone of plasmid pOSCAR. A second NEBuilder HiFi DNA Assembly reaction was then used for the addition to the construct of the *hsv-tk* thymidine kinase gene (from plasmid pGKO2) for selection against ectopic transformants. This recombinant plasmid vector (pOSCAR-hex-KO) was used for transformation of *V. dahliae*, using the *Agrobacterium tumefaciens* strain AGL-1. Monoconidial cultures were prepared from resistant *V. dahliae* colonies, validated as *hex1* deletion mutants by spore PCR and Southern blot analysis (DIG DNA Labeling and Detection Kit, Sigma-Aldrich), and stored as 25% glycerol stocks at −80 °C.

The full-length coding sequence of *hex1* plus 2000 bp long flanks were amplified from the genomic DNA of *V. dahliae* Ls.17 (using the KAPA HiFi DNA polymerase, Roche, Basel, Switzerland) for complementing *hex1* deletion strains. The resulting 4.8 kb long amplicon was co-transformed into *V. dahliae* protoplasts with plasmid pUCATPH, which contains the *hph* cassette (conferring resistance to hygromycin B). Monoconidial transformants were confirmed by spore PCR and stored at −80 °C.

We performed C-terminal *sGFP*-tagging of *hex1* by using the NEBuilder HiFi DNA Assembly kit to generate a construct consisting of the full-length *hex1* open reading frame (ORF) together with its 2000 bp long upstream flank (which presumably includes its endogenous promoter) fused to the *sGFP* coding sequence from plasmid pIGPAPA, the terminal region of *A. nidulans tef1*, and the *hph* selection cassette from plasmid pFC332. The Herculase II Fusion DNA Polymerase was used for the amplification of all fragments, which were then assembled into a new recombinant plasmid (pOSCAR-hex-GFP) in the backbone of pOSCAR by a NEBuilder reaction. This was then transformed into *V. dahliae* Δ*hex1* mutants using the *A. tumefaciens* strain AGL-1. Resistant fungal colonies were single-cell purified, microscopically checked, and stored at −80 °C.

### 2.5. Cytoplasmic and Nuclear Fluorescent Labeling of V. dahliae

The *sGFP* gene was used for fluorescent labeling of either the cytoplasm or histone H1 (as a nuclear label) of *V. dahliae* strains. Construction of strains with cytoplasmic sGFP expression was performed by cloning the *sGFP* expression cassette from plasmid pIGPAPA and the *neo* cassette from plasmid pSD1 to the polylinker of plasmid pBluescript II (using the T4 DNA ligase, Takara Bio, Kusatsu, Japan). The resulting plasmid pBS-GFP-gen was used to transform *V. dahliae* protoplasts, and monoconidial transformants were checked microscopically for robust cytoplasmic sGFP expression.

Plasmid pMF357, which carries the *hph* cassette and a fusion construct of the *sGFP* gene to the *N. crassa* histone H1 gene, was used to transform *V. dahliae* protoplasts for nuclear labeling. Monoconidial transformants were checked microscopically for green nuclear fluorescence.

### 2.6. Morphological and Physiological Characterization of V. dahliae Strains

Previously described methods [34] were used for the morphological and physiological characterization of wild-type and mutant *V. dahliae* strains, with minor modifications. Colony diameters were measured for the determination of growth rates every 5 days during growth on CzD-CM, at 25 °C. Mycelial plugs (1.0 cm in diameter) from the colony periphery of 20 day old cultures were transferred into 500 μL of sterile water and vortexed extensively before determining conidial concentrations using a Neubauer improved cell counting chamber. Frequency of germination was determined by inoculating 1.0 × 10^6^ conidia of each strain into CM and checking microscopically for germling formation after 14 h of growth at 25 °C. All experiments were performed in triplicate.

### 2.7. Plant Pathogenicity Assays

Preparation of eggplant seedlings and conidial suspensions from *V. dahliae* cultures for their pathogenicity assessment was performed as previously described [38], with minor modifications. Seedlings were transplanted at the one true-leaf stage in 100 mL pots containing soil substrate, before drenching with 20 mL of conidial suspension (5.0 × 10^6^ conidia/mL) or sterile water (control plants) per pot. Plants were maintained under controlled conditions at 23 ± 3 °C with a 12 h light/dark cycle.

Typical wilt symptoms were recorded every 3–4 days for 29 days post inoculation (d.p.i.) for monitoring disease progress over time, and at 45 d.p.i. for the determination of final disease severity. Disease parameters were recorded according to previously described criteria [38,39]. Disease severity at each time point was defined as the percentage of symptomatic leaves, and results were plotted over time to generate disease progress curves. These were used for the determination of relative AUDPC (area under disease progress curve) values [38]. In addition, plant growth parameters were recorded at 29 d.p.i. [39].

The presence of the applied *V. dahliae* strain in plant tissues was assessed by fungal re-isolation to verify systemic infection at 29 d.p.i., according to [38]. In particular, nine plant stems per treatment were randomly selected and three xylem chips from different sites along the stem (base, middle, and upper part) were transferred onto acidified PDA medium after the removal of the phloem. The pathogen isolation ratio was expressed as the percentage of positive xylem chips of each plant.

### 2.8. Stress Response Assays and Reactive Oxygen Species (ROS) Detection

Two methods were used to analyze the sensitivity of *V. dahliae* wild-type and mutant strains to varying concentrations of a number of stress-inducing factors and chemicals that are commercially used as fungicides. In summary, we tested growth responses to factors that induce osmotic stress (NaCl, sorbitol), oxidative stress (H_2_O_2_, paraquat, *N*-acetyl cysteine, iprodione), cell wall/plasma-membrane damage or perturbation (calcofluor white M2R, Congo red, SDS, amphotericin B (Biosera, Nuaille, France), fluconazole (Pfizer, Brooklyn, NY, USA), farnesol), and other fungicides that inhibit sporulation and germination (cymoxanil, fosetyl-Al (Bayer, Leverkusen, Germany), mandipropamid, cyflufenamid); chemicals were purchased from Sigma-Aldrich, unless otherwise specified, and fresh solutions were prepared in water directly before each experiment. The first method was a spot dilution assay, in which 10 μL of 10-fold serial dilutions of conidial suspensions (5.0 × 10^6^ to 5.0 × 10^3^ conidia/mL) were spotted on CM agar plates containing the desired stress factor and incubated at 25 °C for 3 days. In addition, for each stress condition/concentration, relative growth inhibition was determined according to the formula: % growth inhibition = ((colony diameter on CM − colony diameter in stress condition)/(colony diameter on CM)) × 100. For each substance considered, a range of concentrations were tested in preliminary experiments for the determination of the minimum inhibitory concentration (MIC) for the wild-type strain, and an appropriate value above the MIC yielding reproducible results between replicates was chosen for further analyses. Each experiment was performed in triplicate.

The generation/accumulation of superoxide anion radicals (O_2_^−^) in the mycelium of *V. dahliae* strains was also examined in response to stress-inducing compounds. For this, the nitro blue tetrazolium chloride (NBT; Cayman Chemical, Ann Arbor, MI, USA) staining method was used; fungal colonies were grown on CM plates supplemented with the desired stress factor for 20 days before the addition of 10 mL of a 0.2% NBT solution on each plate. Following incubation at 25 °C (in the dark) for 45 min, the solution was discarded; then, plates were washed with ethanol, incubated again for 45 min in the dark, and briefly air-dried before recording the assessment of the staining result. All staining assays were performed in triplicate.

### 2.9. Generation of Nit Mutants and Heterokaryon Compatibility Tests

Nitrate non-utilizing (*nit*) mutants of the *V. dahliae* Δ*hex1* strain were generated by the ultraviolet irradiation method and selection on the water agar chloratemedium (WAC), as previously described [40]. Phenotypic classification of *nit* mutants and complementation tests in 96-well plates were also performed according to our previously described procedures [40]. Each pairing was performed in three independent repetitions.

### 2.10. Microscopy

Fungal conidia, germlings, and hyphae were imaged using a Zeiss Axioplan epifluorescence microscope equipped with a differential interference contrast (DIC) optical system, a set of filters G 365 nm (excitation) and LP 420 nm (emission), and a Zeiss Axiocam MRc5 digital camera. Calcofluor white M2R (Sigma-Aldrich) was used for cell wall staining by its addition to the samples (to a final concentration of 10 μg/mL) and incubation at room temperature for 5 min before imaging. Methylene blue staining (final concentration 0.005% *w*/*v*) was used to differentiate live from dead cells. The samples were incubated at 25 °C for 5 min, in the dark, before imaging.

Time-lapse microscopy was performed using a Nikon Ti-E epifluorescence microscope equipped with an autofocus system (Perfect Focus System, Nikon, Minato, Japan), a light-emitting diode (LED) light engine (SpectraX, Lumencor, Beaverton, OR, USA), filter sets 390/18 and 435/48 or 469/35 and 525/50 (excitation and emission filter, respectively; all from Semrock, Rochester, NY, USA except for 525/50, which was from Chroma, Bellows Falls, VT, USA), and a scientific complementary metal–oxide–semiconductor (sCMOS) camera (Flash4, Hamamatsu, Shizuoka, Japan). Images were acquired every 10 min for 24 h (with exposure of 50 and 20 ms for the green and blue channel, respectively), and later processed using ImageJ [41]. Images were adjusted to uniform contrast across all time points, pixel intensity plots were generated using the plot profile (for septae) and surface plot (for the whole area) functions of ImageJ, and background subtraction was performed with a rolling ball radius of 50.0 pixels and default parameters.

### 2.11. Bioinformatic Analyses and Statistics

Homology searches for the identification of Hex1 orthologs in *V. dahliae* and other fungal species were performed using the basic local alignment search tools BlastN, tBlastN, and BlastP searches (NCBI; https://blast.ncbi.nlm.nih.gov/Blast.cgi). All protein sequences were retrieved from GenBank (NCBI; https://www.ncbi.nlm.nih.gov/genbank) and aligned using the MUSCLE algorithm implemented in MEGA X [42]; the alignment was then improved using ClustalW and manually corrected. Reads from previously published RNA-seq experiments of *V. dahliae* were obtained from the Sequence Read Archive (NCBI SRA; https://www.ncbi.nlm.nih.gov/sra; datasets with accession numbers SRP020910, SRP013922, SRP041118, SRP119401, SRP198907, SRP183605, SRP244752, ERP123035, and ERP002524) using BLAST analyses and aligned to the *hex1* coding sequences that flank the predicted intron, using MEGA X.

The maximum likelihood (ML) method, implemented in the MEGA X software suite, was used for the generation of the phylogenetic tree of Hex1 sequences. Sequence alignment was performed using the MUSCLE algorithm, and then it was manually corrected. The Le Gascuel (LG) model, using a discrete Gamma distribution (parameter = 0.4452) with five rate categories, exhibited the lowest Bayesian information criterion (BIC) and corrected Akaike information criterion (AICc) values and was, therefore, chosen as the best-fit substitution model for the construction of the tree. The neighbor-join (NJ) and BioNJ algorithms were applied to generate the initial tree for the heuristic ML search, and the partial deletion option was used for handling alignment gaps. Bootstrap analysis with 1000 replications was performed for assessing tree confidence.

Prediction of the Hex1 secondary structure was performed using the PSIPRED 4.0 Workbench (UCL-CS Bioinformatics, London, UK) [43]. For homology modeling of the Hex1 protein, the Hidden Markov Model-based tool HHPred [44] and MODELLER 9.25 [45] were used, based on the highly similar crystal structure of Hex1 from *N. crassa* (Protein Data Bank, PDB 1KHI) (probability: 99.93%, E-value: 1.9 × 10^−23^). Protein structures were visualized and compared using PyMOL 2.4 (https://pymol.org/2).

Statistical significance of differences between compared datasets was assessed using the analysis of variance (ANOVA) test, following the evaluation of homogeneity of variance across samples (*F*-test *p* ≤ 0.05). Datasets were then subjected to means separation using Tukey’s honestly significant difference (HSD) test.

## 3. Results

### 3.1. Identification and Characterization of the VdHex1 Homolog

To identify homologs of Hex1 in *V. dahliae,* the 176 amino-acid (aa) protein sequence of *N. crassa* (GenBank AAB61278) was used as query in a tBlastN search of the *V. dahliae* genome (strain Ls.17). This yielded a single highly similar sequence of 197 aa (similarity: 80.0%, E-value: 5.0 × 10^−91^), encoded by the 786 bp long gene VDAG_01749. Sequence alignment revealed that this protein was missing the N-terminal hexapeptide MGYYDD, which is, however, highly conserved across Hex1 homologs of various ascomycetes. Further investigation of the gene model VDAG_01749 suggested the presence of a putative intron close to the start codon of the gene that could have been erroneously predicted and led to the description of a truncated ORF. The existence of this intron is conserved among various fungi [17,18,19]. To validate our hypothesis, we retrieved 120 reads from previously generated RNA-seq datasets by BLAST searches against the *hex1* N-terminal region, and we mapped them on its genomic sequence (Figure 1A). More than half of them (67 reads) were mapped on both flanks of our predicted intron, with an internal gap of 217 bp, which corresponds to a genomic sequence with GT–AG boundaries. These findings are consistent with splicing of an intron at these positions, yielding a protein of 188 aa and molecular weight of 21 kDa. Twenty reads exhibited a different mapping pattern that suggests alternative splicing of the intron by using an internal AG donor site (Figure 1A) to produce a larger but highly similar isoform of the Hex1 protein (209 aa, 23.5 kDa). This has also been observed in a number of fungi [18,21]. Finally, 33 reads showed gapless alignments on the genomic sequence of *hex1*, suggesting the presence of pre-messenger RNAs (mRNAs) in the RNA-seq data. Both *V. dahliae* isoforms of Hex1 possess the typical N-terminal hexapeptide MGYYDD and the characteristic peroxisome-targeting signal (SRL tripeptide; PTS1 [16]) at their C-terminus (Appendix A).

Multiple sequence alignment with Hex1 homologs of selected representative species revealed that, apart from the initial hexapeptide, the N-terminal region is highly variable across fungi (Appendix A). In *V. dahliae*, as well as some other species, this region of the protein lacks the polyhistidine motif that has been shown to be involved in Hex1 targeting to the septal wall in *Aspergillus fumigatus* [13]. This suggests that different species have presumably adopted different mechanisms for recruiting WBs to the septum. The remaining part of the protein up to its C-terminus (which constitutes about 78% of the total protein length) is highly conserved, even between species of different orders. Secondary structure prediction revealed that *V. dahliae* Hex1 contains all characteristic motifs, including the 11 beta-sheets and two helices, as well as the conserved histidine residue (at position 51) which has been implicated in crystal formation in *N. crassa* [16] (Appendix A). Furthermore, homology modeling of *V. dahliae* Hex1 demonstrated that the protein is expected to have essentially the same tertiary structure as the solved structure of its *N. crassa* homolog (PDB 1KHI [16]), notably with two perpendicular antiparallel β-barrels, except for the variable N-terminal region, which is probably in a disordered state (Figure 1B).

Genomic searches revealed that all known *Verticillium* species possess a single Hex1 homolog, with the exception of *V. longisporum*, a nearly diploid hybrid [46], which has two nonidentical paralogs. Phylogenetic analysis of the Hex1 homologs of *Verticillium* and members of several Pezizomycotina orders of varying phylogenetic distances demonstrated that the evolution of this single-copy gene generally follows the divergence pattern of the species (Figure 1C).

### 3.2. Deletion of Vdhex1 and Morphological and Physiological Characterization

To determine the functions of Hex1 in *V. dahliae*, a deletion mutant (Δ*hex1*) was constructed through double homologous recombination using *Agrobacterium tumefaciens* for transforming wild-type conidia (isolate Ls.17). Replacement of the wild-type *hex1* gene with the *neo* cassette (conferring resistance to geneticin) was confirmed using PCR and Southern blot analyses (Appendix A). The full-length *hex1* ORF of the wild-type isolate, flanked by 2 kb long genomic sequences, was amplified and co-transformed with plasmid pUCATPH to generate the complemented strain *hex1*-c.

When compared to the wild type and the complemented strain, the Δ*hex1* mutant exhibited significantly slower growth (*p*-value: 9.0 × 10^−4^), more compact aerial mycelium, and abnormal colony periphery, both on PDA and on MM (Figure 2A,B). Microsclerotia and pigment deposition in the knockout strain were delayed by 5 days; however, after 30 days of growth, they were indistinguishable from the other strains. The mutant exhibited significant phenotypic alterations regarding its asexual reproductive capacity. Conidiation was dramatically lower than in the wild type (10-fold reduction, *p*-value: 8.7 × 10^−7^; Figure 2B), and the ability of its conidia to germinate was also significantly reduced (15.3% reduction, *p*-value: 1.8 × 10^−5^) in comparison to the wild type (Figure 2B). Furthermore, we frequently observed abnormal morphologies in Δ*hex1* germlings, which included bipolar or irregular patterns of germ tube emergence, and faster branching of young germlings (Figure 2C). Mature Δ*hex1* hyphae and conidiophores also exhibited a distinctive “curly” phenotype in their apical regions, which often appeared wavy and fragile (Figure 2D; Appendix A). All aforementioned phenotypes were fully rescued by complementation of *hex1* in the *hex1*-c strain, confirming their ascription to the *hex1* deletion. On the other hand, the mutant’s septation pattern was not significantly altered (Figure 2D).

The three strains were subjected to hypotonic shock by immersing their hyphae into distilled water, in order to study their integrity and resilience. Notably, hyphal burst and cytoplasmic leakage were frequently observed only in the Δ*hex1* strain, mostly at the hyphal tips and rarely at subapical compartments, leading sometimes to shrunk hyphae that had lost their cytoplasm (Figure 2E). During live-cell imaging of hyphae of both the wild-type and the rescued strains, we also often observed spherical vesicles in the vicinity of septae (and rarely close to the cell wall without obvious connection to septae), which were not detected in Δ*hex1*. Overall, these results are consistent with the hypothesis that Hex1 in *V. dahliae* is essential for WB formation, which is involved in the maintenance of hyphal integrity, presumably by conditionally sealing the septal pore.

### 3.3. Localization of VdHex1 at Septae

Our finding that the *V. dahliae* Hex1 homolog lacks a polyhistidine motif in its N-terminal region, which has been demonstrated to be involved in septal targeting in *A. fumigatus* [13], motivated us to examine directly its subcellular localization. For this, we constructed a *V. dahliae* strain expressing a Hex1-sGFP construct under the control of its endogenous promoter. Fusion of sGFP at the C-terminus of Hex1 was expected to largely inactivate the terminal peroxisome signal motif [47]. Indeed, vesicles with sGFP signal, presumably corresponding to peroxisomes or WBs, were only relatively rarely observed (Figure 2F). On the other hand, strong fluorescence was often detected at septae; notably, this was not limited to septal pores, but was distributed along the entire septum (Figure 2F). These results directly demonstrate the association of Hex1 with septae in *V. dahliae* and suggest that different species have probably adopted different mechanisms for targeting Hex1 to the septum.

### 3.4. Vdhex1 Is Indispensable for Pathogenicity

Pathogenicity bioassays on eggplant (cv. Black Beauty), which is highly susceptible to *V. dahliae* [38], were performed to investigate the possible involvement of Hex1 in pathogenicity and virulence of the fungus. Eggplant seedlings were inoculated with the wild-type, Δ*hex1*, and *hex1*-c strains, and their disease severity was assessed 45 days after inoculation. The Δ*hex1* mutant caused notably less severe disease compared to the wild-type and *hex1*-c strains (Figure 3A,B). Only a small percentage of Δ*hex1*-inoculated plants (6.7%) showed severe symptoms (disease severity >50%), while the majority (86.7%) remained symptom-free or exhibited very mild symptoms (disease severity ≤20%). This was in contrast to the wild type and the complemented strains, which caused severe symptoms to 36.7% and 30.0% of plants, respectively, characterized by typical *Verticillium*-induced symptoms including wilting, chlorosis, and defoliation (Figure 3B).

To gain a better understanding of the pathogenic properties of the Δ*hex1* deletion mutant, we performed an additional detailed time-course comparison of virulence using pathogenicity assays of the wild type and Δ*hex1* strains over 29 days post inoculation (Figure 3C). Plants inoculated with the wild type started presenting symptoms on day 15, and the mean disease severity at the end of the experiment was 59.4% (±4.7%). On the contrary, infection with the Δ*hex1* mutant started causing symptoms only on day 22, reaching a final severity of 9.1% (±3.3%). Calculation of the area under disease progress curve (AUDPC) values, which account for the plant disease progression over 29 days, revealed that the Δ*hex1* virulence was 16.3 times lower than that of the wild type (Figure 3D). Consistently, the average fresh weight of plants treated with Δ*hex1* was significantly higher than that of plants treated with the wild type (Figure 3E). Finally, we investigated the presence of the fungus within plant tissues by attempting to re-isolate it from xylem sections of the infected plants, at the end of the experiment. The re-isolation ratio of Δ*hex1* was reduced by 49.4% in comparison to the wild type (0.41 ± 0.09 and 0.81 ± 0.08, respectively; Figure 3F), which corroborates the difference observed in disease severity and demonstrates the difficulty of the Δ*hex1* mutant to colonize the xylem and cause systemic infection of the plants.

### 3.5. VdHex1 Is Involved in the Fungal Response to Osmotic Stress and Cell Wall-/Plasma Membrane-Perturbating Agents

On the basis of our previous findings and the demonstrated functions of Hex1 and WBs in other fungi [27], we supposed that the Hex1 homolog of *V. dahliae* might also be involved in environmental stress responses and the maintenance of cellular integrity. To test this, we exposed the deletion mutant Δ*hex1* to hyperosmotic stress and a variety of cell wall-perturbating or damaging substances, and we compared its responses to those of the wild-type and the complemented strains by spot dilution assays and radial growth analyses. Upon treatment with either NaCl or sorbitol, the Δ*hex1* strain achieved remarkably reduced growth in comparison to the other strains, even though germination as revealed by our spot assays was not significantly affected (Figure 4A,B). The observed growth inhibition appeared dependent on the substance used (i.e., exposure to 0.5 M NaCl caused the same level of inhibition as 1.0 M sorbitol, Figure 4B), while our preliminary tests demonstrated that the extent of inhibition was also concentration-dependent. Microscopic observation of the three strains after growth in a hyperosmotic medium of 0.5 M NaCl revealed that many Δ*hex1* germlings exhibited hyphal bleeding of their cytoplasm, and the majority of them were entirely permeable to methylene blue, which stains dead cells. On the contrary, wild-type and rescued cells remained mostly unstained, or only one compartment allowed accumulation of the dye, while the remaining hypha was unaffected (Figure 4C). This suggests that Hex1 is involved in hyphal compartmentalization and ensures cell heterogeneity, presumably by occluding septal pores upon osmotic stress to protect the remaining hypha from destruction. Furthermore, hyperosmotic conditions induced increased septation in Δ*hex1*, as well as increased the frequency of “curly” hyphae with branching and the appearance of “bubble”-like cells, only in the mutant (Figure 4C). These findings are further suggesting increased sensitivity of Δ*hex1* cells to osmotic stress.

Calcofluor white, Congo red, and SDS were used to test the sensitivity of the Δ*hex1* mutant to cell wall- and membrane-damaging agents. Germination frequency of the mutant was slightly reduced in the presence of Congo red, and its radial growth was significantly limited upon exposure to calcofluor white and Congo red, suggesting reduced tolerance against cell wall stress (Figure 5A,B). It should be noted that the sensitivity of the Δ*hex1* mutant was remarkably higher to calcofluor white in comparison to Congo red (Figure 5A,B). Similar effects were observed when we tested the antifungal agents amphotericin B and the acyclic sesquiterpene alcohol farnesol, which target sterol biosynthesis and membrane integrity (Figure 5A,B). Fluconazole was also tested but seemed to affect equally the deletion mutant and the control strains (Appendix A).

Finally, we tested a number of substances that are present in commercially available fungicides as inhibitors of fungal sporulation and germination (cymoxanil, fosetyl-AI, mandipropamid, and cyflufenamid). All of them affected similarly the mutant and the control strains (Appendix A).

### 3.6. VdHex1 Is Required for Normal Response to Oxidative Stress and ROS Metabolism

Understanding of the strategies that organisms have evolved to cope with oxidative stress, on one hand, and to use reactive oxygen species (ROS) as important signaling components in developmental regulation, on the other, are of fundamental interest; this becomes especially relevant in the case of pathogens, which need efficient mechanisms to counteract host-induced oxidative responses [48,49]. Since the role of Hex1 in response to oxidative stress is not well established, we used the oxidative agents H_2_O_2_, the herbicide paraquat, and iprodione (i.e., a broad-spectrum antifungal drug that blocks germination and causes oxidative damage), as well as *N*-acetyl cysteine as ROS scavenger, to examine the possible involvement of the protein in the resistance of the fungus to ROS. Treatment with any of the three ROS-inducing substances resulted in growth inhibition of the Δ*hex1* mutant and induced pigmentation only in the wild-type and complemented strains, while iprodione additionally caused a drastic reduction in germination frequency, particularly in Δ*hex1* (Figure 6A,B). We note that, in the case of H_2_O_2_, significant differences between the null mutant and the other strains were only observed in intermediate concentrations (Figure 6A,B), while the same trend but lacking statistical significance was observed when concentrations lower than 0.7 mM and higher than 2.0 mM were tested.

To further investigate how Hex1 possibly affects ROS levels under different types and levels of stress, we used a histochemical method of NBT staining to detect superoxide radicals (O_2_^−^) in the mycelium. We indeed observed a different behavior regarding ROS accumulation between Δ*hex1* and the control strains. Under standard conditions (CM), as well as upon treatment with the stressors paraquat and farnesol, the mutant displayed more limited ROS levels, restricted to a narrow peripheral mycelial area of active hyphal growth (Figure 6C). In sharp contrast, exposure to hyperosmotic stress (NaCl or sorbitol) resulted in drastically elevated ROS levels throughout the colony. These findings suggest involvement of Hex1 in the regulation of ROS metabolism as a function of environmental stress.

### 3.7. VdHex1 Is Not Involved in Heterokaryon Incompatibility

Vegetative fusion of hyphae from different individuals to the formation of heterokaryons is strictly regulated by heterokaryon incompatibility systems. If the two fusion partners have genetic differences in their *het* loci, the heterokaryotic cell compartment is isolated from the rest of the mycelium and often destroyed through a targeted cell death reaction [7]. On the basis of our findings, which suggest that Hex1 and WBs of *V. dahliae* are involved in hyphal compartmentalization, presumably by conditionally sealing septal pores upon mechanical damage or environmental stress, we hypothesized that they might also be responsible for the isolation of heterokaryotic cells when incompatibility reactions are triggered. To investigate this possibility, we first checked whether *hex1* in *V. dahliae* is required for hyphal fusion or involved in the control or activation of the heterokaryon incompatibility reaction per se. For this, we used the Δ*hex1* deletion mutant in compatibility assays with standard tester strains from all known compatibility groups of *V. dahliae* (VCGs). In these tests, complementary nitrate non-utilizing (*nit*) mutants of tested isolates are used in pairings on medium which would support only heterokaryotic growth. Therefore, *nit1* and *nitM* mutants of the Δ*hex1* strain of isolate Ls.17 were generated and used in assays with complementary *nit* mutants of 23 wild-type isolates of all compatibility groups; self-pairings (i.e., with independent *nit* mutants of the same wild-type isolate, Ls.17) were also performed as controls (Appendix A). The results are summarized in Appendix A. Mutants of the Δ*hex1* strain were compatible with each other and gave rise to robust heterokaryons, suggesting that *hex1* is not required for hyphal fusion. Mutants of both the wild-type isolate Ls.17 and its Δ*hex1* descendant exhibited strong reactions with members of the same compatibility group, as expected, as well as weaker or inconsistent interactions with members of other groups, which is often observed in this species [40]. No difference in the compatibility profile of the Δ*hex1* mutant was recorded, suggesting that the gene is not involved in the genetic control of heterokaryon incompatibility.

We then addressed our hypothesis that Hex1 might be involved in the containment of the incompatibility-induced catastrophic reaction to the heterokaryotic cell, using time-lapse live-cell imaging of fusion events between incompatible strains. We chose the wild-type isolate BB, which is incompatible with Ls.17, and we also deleted its *hex1* gene, using the same strategy as for our reference strain (Appendix A). In addition, we labeled the nuclei of BB by tagging its histone H1 with sGFP and the cytoplasm of Ls.17 with cytoplasmically localized sGFP. This would permit the direct microscopic identification of the two strains in pairings and detection of fusion events between the two. Using this strategy and the two engineered strains, we detected eight independent fusion events between the two incompatible strains and recorded time-lapse movies (24 h) to study the cell fate upon fusion. In all cases, fusion was followed by gradual nuclear degradation and cell shrinkage, which are typical manifestations of heterokaryon incompatibility-induced cell death [7] (Figure 7A). These processes were always confined to the heterokaryotic cell, and they never affected the adjacent cell compartments (Figure 7A; Appendix A). These results suggest that WBs and their major component, Hex1, are not necessary for septal pore plugging in response to the activation of heterokaryon incompatibility cell death reactions. On the other hand, in all of these cases, staining with calcofluor white indicated highly increased chitin deposition on the septae which surround the heterokaryotic cell that is destroyed (Figure 7A,B; Appendix A), as well as on the cell wall of the compartment (Appendix A). Although it is not clear whether this has a role in the reaction or is just a consequence, a reasonable hypothesis is that this extreme thickening of septae might be ensuring the efficient sealing of the heterokaryotic cell and preventing the diffusion of cell death mediators to the adjacent cells.

Finally, according to the established functions of WBs in controlling traffic through the septal pore of fungi, we contemplated that they could also be involved in the control of nuclear migration, an essential step to the formation of heterokaryons through hyphal fusions. However, microscopic examination of the BB Δ*hex1* strain with sGFP-labeled nuclei revealed that essentially all hyphal compartments were strictly uninucleate (Appendix A), with the exception of the actively dividing cells at hyphal tips, which can transiently have more than one nuclei until the formation of the new septum. This was also the case in the wild-type strain. We rarely observed nuclei migrating through the septal pore, both in wild-type and in Δ*hex1* hyphae (Appendix A). A possible explanation for the fact that binucleate cells are, nevertheless, never observed in the mycelium could involve degradation of one of the nuclei upon migration [50].

## 4. Discussion

Multicellularity has independently evolved from unicellular ancestors in different eukaryotic lineages, with diverse outcomes [51]. In filamentous fungi, these transitions led to the evolution of the syncytial mycelium, which is characterized by cytoplasmic continuity [52]. This ensures efficient translocation of nutrients and genetic resources [53] and coordination of responses to environmental changes [54], without the need for a dedicated vascular system. Nevertheless, uncontrolled intercompartmental traffic would lead to increased exposure to biotic and abiotic risks that could rapidly propagate throughout the mycelium [3,4,5,6,7], which underlines the necessity for a trade-off between continuity and conditional control of exchanges. Fungal evolution has invented structures and mechanisms that address this need, including septation of hyphae in higher fungi (ascomycetes and basidiomycetes) [2] and regulated mechanisms for the control of cytoplasmic flow through septal pores [8]. In ascomycetes, the peroxisome-derived WBs are involved in this process of plugging septal pores as a response to mechanical or other stimuli [9,27,55].

The major component of WBs was shown to be the conserved peroxisomal protein Hex1 in *N. crassa* [15,17]. Its discovery was followed by detailed structural and functional analyses, mostly in this and to a lesser extent in other fungi [13,16,18,20,26]. In this study, we identified the single Hex1 homolog of *V. dahliae*, a plant pathogen with a particularly broad range of hosts and significant economic impact [28]. A careful study of the deposited gene model, using automatic annotation, led us to revise the exon–intron boundaries of the gene and further revealed that it is subject to alternative splicing, a process that appears to be very common in *V. dahliae* [56] and which, specifically for *hex1*, has also been reported in certain other fungi [17,18,19]. The expression of two protein isoforms has also been experimentally demonstrated in other species using Hex1-specific antibodies [17,18,21], which corroborates our suggestion. The inaccuracy of automatic gene annotation in this case might explain the predicted hypervariable N-terminal extensions observed in many sequences from different species [25,26], indicating that these data should be treated with caution.

Woronin bodies are peroxisomal vesicles with a dense core of self-assembled Hex1 into a crystal lattice [16]. Both *V. dahliae* isoforms of Hex1 were predicted to adopt essentially the same tertiary structure as their *N. crassa* homolog [16], with the exception of the N-terminal region, which is extremely variable in sequence between fungi and seems to remain in a disordered state. Typical peroxisomal targeting signal peptides [47] were detected at their C-termini, and the conserved histidine residue that is necessary for crystallization of the protein is also present [16]. None of the isoforms have a polyhistidine motif in the variable N-terminal region, which in *A. fumigatus* is necessary for septal targeting of Hex1 [13]. However, such motifs are also absent from other Hex1 homologs, including that of *N. crassa* [17]. Moreover, C-terminal tagging of the *V. dahliae* Hex1 with sGFP demonstrated that Hex1 is still targeted to the septum, despite the lack of this motif. Overall, these findings suggest that more than one mechanism mediates septal targeting of Hex1 in fungi. On the basis of this localization pattern and the well-established role of Hex1 in septal pore plugging, we investigated the resilience of mutant Δ*hex1* hyphae. In full concordance with its expected function, the lack of Hex1 rendered *V. dahliae* hyphae prone to excessive hyphal bleeding upon hypotonic shock-induced hyphal damage. This suggests defects in hyphal compartmentalization due to dysfunctional septal pore sealing.

Growing evidence from various fungi indicates that, apart from its defining role in septal plugging upon damage, Hex1 has pleiotropic phenotypes in fungal morphology and physiology [13,17,18,22,26]. Some of these reports present contradictory results regarding particular phenotypes in different species, including growth characteristics and pathogenicity. These considerations served as our motivation to investigate a number of hypothesized roles of Hex1 in physiology, pathogenicity, stress response, ROS metabolism, and heterokaryon incompatibility in *V. dahliae*. Our results suggest the multifaceted involvement of Hex1 in several fundamental processes in this fungus. Deletion of *hex1* led to reduced growth rate, similarly to *N. crassa* [15,17] and *Arthrobotrys oligospora* [24]*,* unlike several other fungi, in which deletion of the gene did not result in growth defects [18,21,22,25]. The absence of Hex1 from *V. dahliae* also resulted in drastic reduction in conidiogenesis, which has been generally observed [17,22,24,25]. Furthermore, we identified in *V. dahliae* developmental phenotypes that were hitherto unknown for this gene, namely, restricted and abnormal conidial germination, increased hyphal branching, “curly” hyphal tips, and “bubble”-like cells. Collectively, *V. dahliae* Hex1 appears to be involved in a number of fundamental processes that are related to asexual reproduction, hyphal development, and possibly cell wall deposition and cellular polarity. An enrichment of WBs in the apical region of germ tubes and hyphal tips has been described in *N. crassa* and *A. nidulans* [11,57,58], and their peroxisome-unassociated apical clustering has been shown in *A. fumigatus* [21]. These resemble the localization pattern of the Spitzenkörper, the organizing center of hyphal growth that coordinates the vesicle-mediated delivery of cell wall material to the apical cell surface [59]. These observations and the pleiotropic growth phenotypes that we report here open up the possibility that Hex1 and WBs might have multiple cellular roles in growth and morphogenesis of *V. dahliae* and other fungi, some of which might be independent from their established role in septal pore occlusion upon damage, a hypothesis that welcomes future investigation.

Motivated by the identification of phenotypes in Δ*hex1* cells that suggest reduced cell wall robustness, including the detection of hyphal bleeding from internal compartments upon hypotonic shock and the “curly” hyphal tips that were often observed, we further investigated the responses of the mutant strain to osmotic stress and to factors that affect cell wall and membrane integrity. We found that, apart from growth inhibition, presumably due to the reduced ability of the hyphae to isolate their damaged cellular compartments, hyperosmotic conditions also caused increased septation and frequent appearance of morphological alterations, further supporting the hypothesis that Hex1 contributes to the regulation of colony morphogenesis. Moreover, our experiments demonstrated that Hex1 in *V. dahliae* mediates resistance against common fungicides that target the cell wall and the plasma membrane. In the absence of Hex1, we observed reduced germination of conidia and growth rate upon treatment with either calcofluor white or Congo red, which indicates that changes in cell wall structure and related defects in hyphal integrity possibly result in impaired germination. Contrasting results have been reported with regard to the sensitivity of Δ*hex1* mutant strains of other fungi to osmotic and cell wall/plasma-membrane stress; deletion of the gene did not significantly affect *M. robertsii* and *A. flavus* [25,26], whereas, in *A. fumigatus* and *Arthrobotrys oligospora*, it had a significant impact [13,24].

We further hypothesized and investigated whether Hex1 could be involved in the fungal response to oxidative stress and ROS metabolism. The mutant strain indeed exhibited growth defects in coping with increased oxidative damage induced by treatment with H_2_O_2_ and paraquat, as well as with amphotericin B, farnesol, and iprodione, but not fluconazole. This may indicate that its hyphal development might be affected not only by inhibition of ergosterol biosynthesis, but also by the oxidative damage that these factors can cause. Furthermore, NBT staining of superoxide in the mycelium indicated altered levels of ROS accumulation, both under standard conditions and in the presence of high oxidative load. Osmotic stress caused increased ROS accumulation in the Δ*hex1* mycelia and this could possibly explain the significant developmental defects in these conditions. On the other hand, in the presence of farnesol, Δ*hex1* exhibited lower ROS generation. These findings open up the possibility that Hex1 plays a role in ROS metabolism and resistance to oxidative stress in *V. dahliae*. Considering the significance of intracellular ROS in signaling and regulatory functions [60], such a connection with Hex1 might be worth investigating further. Furthermore, it possibly becomes especially relevant in pathogenic fungi, which need to cope with host responses to infection that often involve bursts of ROS generation [48,49]. In the pathogenic fungi *Alternaria brassicicola* and *A. fumigatus*, WBs were shown to be associated with the redox homeostasis-related transmembrane protein TmpL [61], which could be consistent with a function of Hex1 in ROS metabolism. To our knowledge, we report here the first experimental data that suggest involvement of Hex1 in ROS homeostasis.

Considering the pleiotropic and significant effects that Hex1 deletion has on fungal growth and development, one could reasonably further assume a possibly important role in pathogenicity. This has been investigated in several plant and other pathogens, and results between species are contradicting, demonstrating partial attenuation of virulence to varying degrees in some cases [13,18,22,25], but no compromise in pathogenic capacity in others [23,26]. We report here, for *V. dahliae*, one of the most pronounced roles of Hex1 in fungal pathogenicity. Deletion of *hex1* diminished virulence of *V. dahliae* on eggplant, a very susceptible host of this pathogen [38]. Consistently with this drastically reduced disease severity (by 84.7% on average), presence of the fungus in the xylem of infected plants was also significantly reduced (by 49.4%). These results suggest that Hex1 is required not only for penetration of roots by *V. dahliae*, but also for its propagation in the host and its ability to cause systemic infection. It is reasonable to assume that this drastic reduction of virulence could be linked to the pleiotropic defects of Δ*hex1* in *V. dahliae* growth, physiology, and stress response, rather than to solely attribute the attenuated virulence to an independent role of Hex1 in pathogenicity. The infection cycle of *V. dahliae* begins with germination of microsclerotia, elongation of hyphae, and penetration of roots. In this endeavor, hyphal integrity is indispensable for efficient attack to the plant [62]. We demonstrated that deletion of Hex1 negatively affected germination and hyphal growth, as well as cell wall integrity and resistance to stress conditions. Furthermore, propagation of the fungus in plant tissues, mostly within xylem vessels, is required in subsequent stages of the plant disease, which involves sequential conidiation, germination, and hyphal growth cycles [63]. However, we found that the Hex1-deficient mutant has a diminished ability to produce conidia, which would probably challenge its ability to efficiently colonize the infected plant [64]. Finally, it is possible that the lack of Hex1 and functional WBs might negatively interfere with the ability of the fungus to withstand the stressful adverse conditions induced in the xylem by the plant’s defense mechanisms. For example, one of the host responses to fungal invasion involves rapid bursts of ROS [48]. Since Δ*hex1* cells are defective in their responses to oxidative stress, this could be an additional factor that compromises its pathogenic potential.

Establishment of viable heterokaryons in fungi, resulting from hyphal fusion between different individuals, is often prevented in natural populations by incompatibility mechanisms. One of them involves the triggering of a cell death reaction if the fused cells are genetically incompatible, resulting in intense vacuolization, nuclear degradation, cell wall shrinkage, and, finally, cell death [7,33]. However, this reaction is highly localized and only affects the fused cells, whereas the adjacent hyphal compartments remain unaffected [7]. We hypothesized that the mechanism that is responsible for sealing of the incompatible fused cells to prevent the diffusion of cell death mediators to neighboring cells might involve the WBs, in a function analogous to septal pore plugging upon mechanical damage. However, our time-lapse imaging experiments of incompatible fusions clearly demonstrated that this compartmentalization function is independent of Hex1, as it normally occurs in Δ*hex1* hyphae. This is similar to what was previously observed in *N. crassa* [65]. In the course of our experiments, we invariably observed the fusion-induced thickening of septae defining the boundaries of the fused incompatible cells, presumably by intense chitin deposition. This was also detected, to a lesser extent, in the cell wall of these cells, but never affected any of the adjacent hyphal compartments. Similar observations were previously made in *Podospora anserina* [66]. We propose that the reinforcement of the septae of incompatible cells is involved in their sealing mechanism, which aims at preventing the incompatibility reaction from spreading in the hyphae, as well as protecting the neighboring cells once their fused neighbor gets disorganized as the result of the incompatibility reaction.

In conclusion, we demonstrate here important and pleiotropic roles of the protein Hex1 in the plant pathogenic fungus *V. dahliae*. Apart from its function in hyphal compartmentalization in response to hyphal damage, involvement of the protein was identified in fundamental biological processes related to fungal growth, physiology, asexual reproduction, stress response, and pathogenicity. Although Hex1 is highly conserved in ascomycetes, diverse functions have been attributed to its homologs in different species, which underlines the necessity of further functional investigations to fully elucidate its multifaceted role in fungal biology.

## Figures and Tables

**Figure 1 jof-06-00344-f001:**
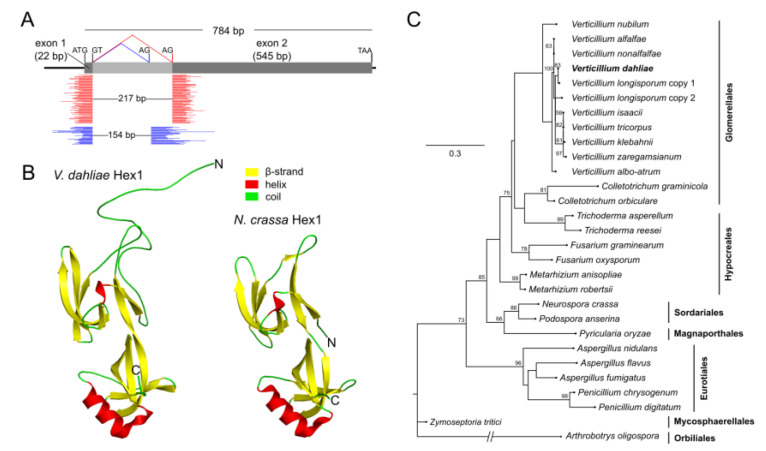
Gene and protein structure of the Hex1 homolog of *Verticillium dahliae*, and phylogenetic analysis. (**A**) Gene structure of locus VDAG_01749. Mapping of RNA-seq reads on the genomic sequence supports the definition of an intron that can be spliced in two alternative ways (red and blue reads, respectively) to produce two highly similar isoforms of the protein. (**B**) Prediction by homology modeling of the tertiary structure of *V. dahliae* Hex1 and comparison to the solved structure of its *Neurospora crassa* homolog (PDB 1KHI). (**C**) Maximum likelihood phylogenetic tree of Hex1 homologs across several Pezizomycotina orders of varying phylogenetic distances from *Verticillium*.

**Figure 2 jof-06-00344-f002:**
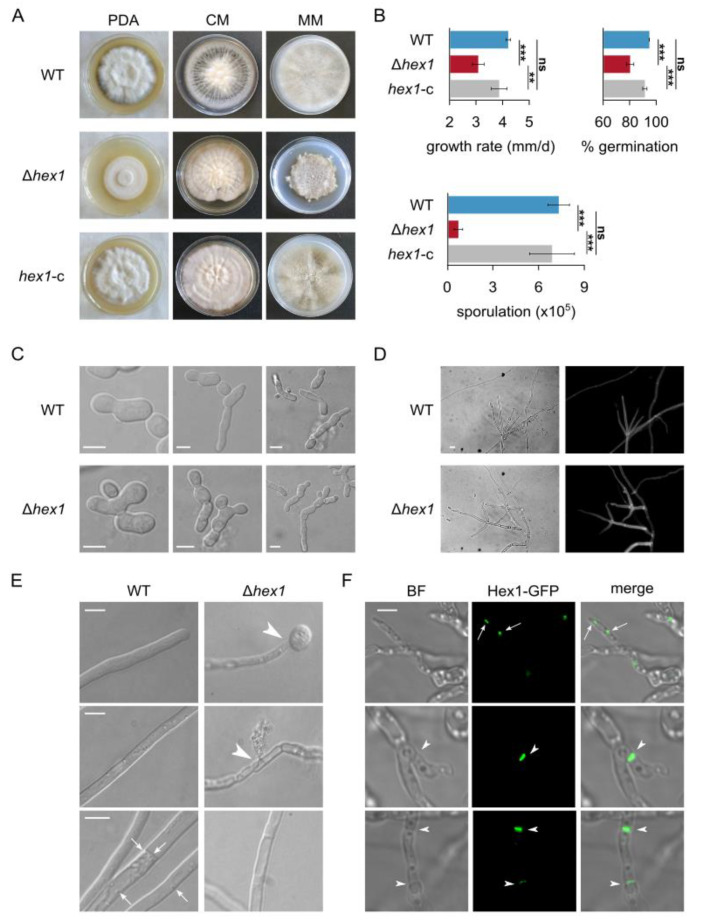
Morphological and physiological characterization of the *V. dahliae* Δ*hex1* strain. (**A**) Colony morphologies of the *V. dahliae* wild-type isolate (Ls.17), the deletion mutant Δ*hex1*, and the complemented strain *hex1*-c, after growth for 25 days on PDA, CzD-CM, and MM. CM: CzD-CM. (**B**) Growth rate, ability of conidia to germinate, and abundance of conidial production of the three strains. All experiments were performed in triplicate, and 150 conidia were tested for germination per replicate. Bars: SD. Statistical significance of differences was tested by one-way ANOVA, followed by Tukey’s post hoc test (** *p* ≤ 0.01, *** *p* ≤ 0.001, ns: nonsignificant). (**C**,**D**) Microscopic characteristics of conidia (**C**; bars = 5 μm) and hyphae (**D**; cell wall staining using calcofluor white M2R; bar = 10 μm) of the wild-type and the Δ*hex1* strains. (**E**) Response of the wild-type and the Δ*hex1* strains to hypotonic shock by immersing their hyphae to distilled water. Arrowheads: hyphal burst and cytoplasmic leakage. Arrows: spherical vesicles, usually localized close to the septal wall, that were observed during live-cell imaging of the wild-type strain but were absent from Δ*hex1*. Bars = 10 μm. (**F**) Subcellular localization of the Hex1-sGFP tagged protein in *V. dahliae*. Hex1 is localized either in small globular vesicles (presumably peroxisomes or Woronin bodies (WBs), indicated by arrows) or at the septal wall (arrowheads). Bar = 5 μM.

**Figure 3 jof-06-00344-f003:**
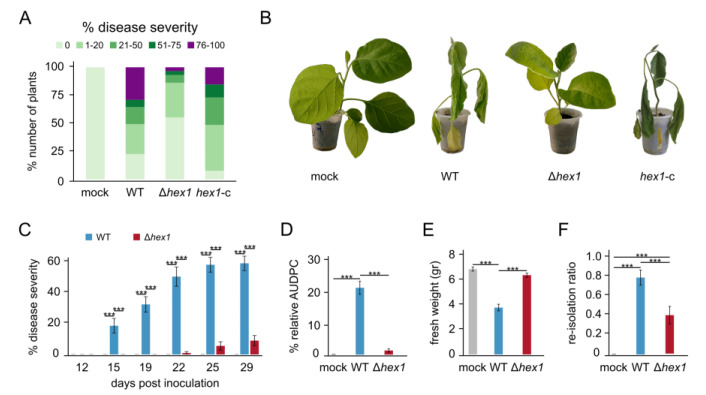
Phytopathological characterization of the *V. dahliae* Δ*hex1* strain. (**A**) Disease severity of the wild-type (WT), the deletion (Δ*hex1*), and the complemented (*hex1*-c) strains, 45 days post inoculation on eggplant seedlings (30 plants/strain). Uninfected plants (mock) served as control. (**B**) Representative plants are shown for each strain. (**C**) Time-course comparison of disease severity between the wild type and Δ*hex1* (21 plants/strain) over 29 days post inoculation. (**D**–**F**) Mean relative area under disease progress curve (AUDPC) scores, plant fresh weights, and fungal re-isolation ratios at the end of the time-course experiment. Bars = standard error (SE). Statistical testing by one-way ANOVA followed by Tukey’s post hoc test (*** *p* ≤ 0.001).

**Figure 4 jof-06-00344-f004:**
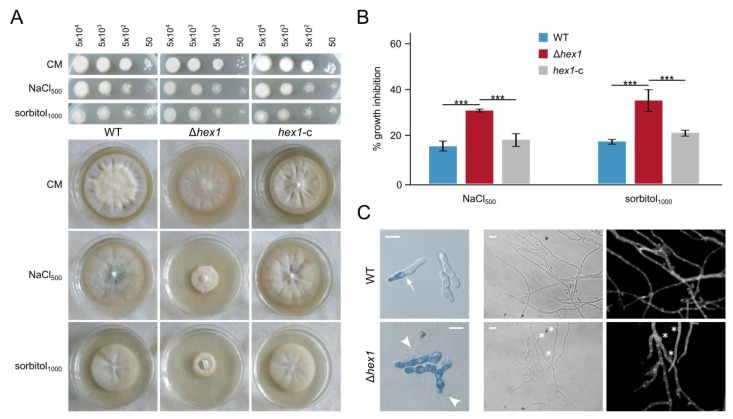
Effects of hyperosmotic stress on Δ*hex1*. (**A**) Responses of the wild-type, the deletion (Δ*hex1*), and the complemented (*hex1*-c) strains to NaCl and sorbitol, with regard to germination (top; the number of inoculated conidia per spot are provided above the images; growth for 3 days) and radial growth (bottom; growth for 18 days). CM: CzD-CM. (**B**) Relative growth inhibition by NaCl and sorbitol (each condition was tested in triplicate; bars = SD; statistical testing by one-way ANOVA followed by Tukey’s post hoc test (*** *p* ≤ 0.001). All concentrations are given in mM. (**C**) Staining of wild-type and Δ*hex1* germlings with methylene blue, after growth for 16 h in a hyperosmotic medium of 0.5 M NaCl (left). Arrow: septum. Arrowheads: cytoplasmic bleeding. Morphology of corresponding hyphae under the same conditions. Asterisks: “bubble”-like cells. Cell wall staining using calcofluor white M2R. Bar = 10 μm.

**Figure 5 jof-06-00344-f005:**
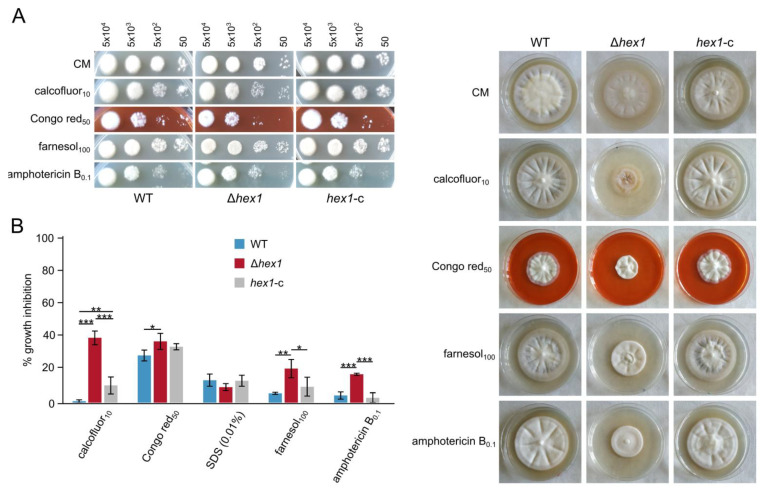
Effects of cell wall and plasma-membrane stress on Δ*hex1*. (**A**) Responses of the wild-type, the deletion (Δ*hex1*), and the complemented (*hex1*-c) strains to calcofluor white M2R, Congo red, SDS, farnesol, and amphotericin B with regard to germination (top; the number of inoculated conidia per spot are provided above the images; growth for 3 days) and radial growth (right; growth for 18 days). CM: CzD-CM. (**B**) Relative growth inhibition by the same substances (each condition was tested in triplicate; bars = SD; statistical testing by one-way ANOVA followed by Tukey’s post hoc test (* *p* ≤ 0.05, ** *p* ≤ 0.01, *** *p* ≤ 0.001). All concentrations are given in μg/mL.

**Figure 6 jof-06-00344-f006:**
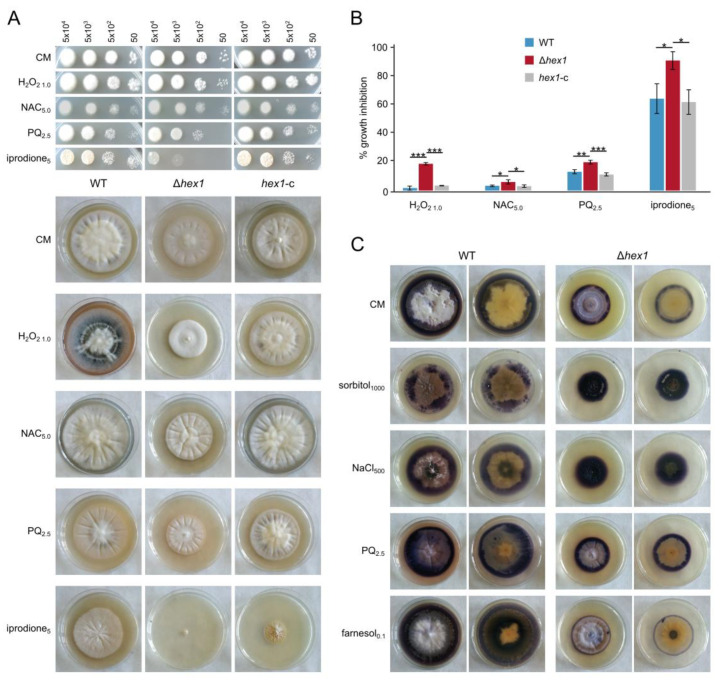
Effects of oxidative stress on Δ*hex1* and its involvement in reactive oxygen species (ROS) metabolism. (**A**) Responses of the wild-type, the deletion (Δ*hex1*), and the complemented (*hex1*-c) strains to H_2_O_2_, paraquat, iprodione, and *N*-acetyl cysteine with regard to germination (top; the number of inoculated conidia per spot are provided above the images; growth for 3 days) and radial growth (bottom; growth for 18 days). CM: CzD-CM. (**B**) Relative growth inhibition by the same substances (each condition was tested in triplicate; bars = SD; statistical testing by one-way ANOVA followed by Tukey’s post hoc test (* *p* ≤ 0.05, ** *p* ≤ 0.01, *** *p* ≤ 0.001). All concentrations are given in mM, except for iprodione (μg/mL). (**C**) Detection of superoxide anion radicals (O_2_^−^) using nitro blue tetrazolium chloride (NBT) staining in colonies of the two strains upon treatment with various sources of stress (NaCl, sorbitol, paraquat, and farnesol). Both top and bottom views of plates are shown for every treatment.

**Figure 7 jof-06-00344-f007:**
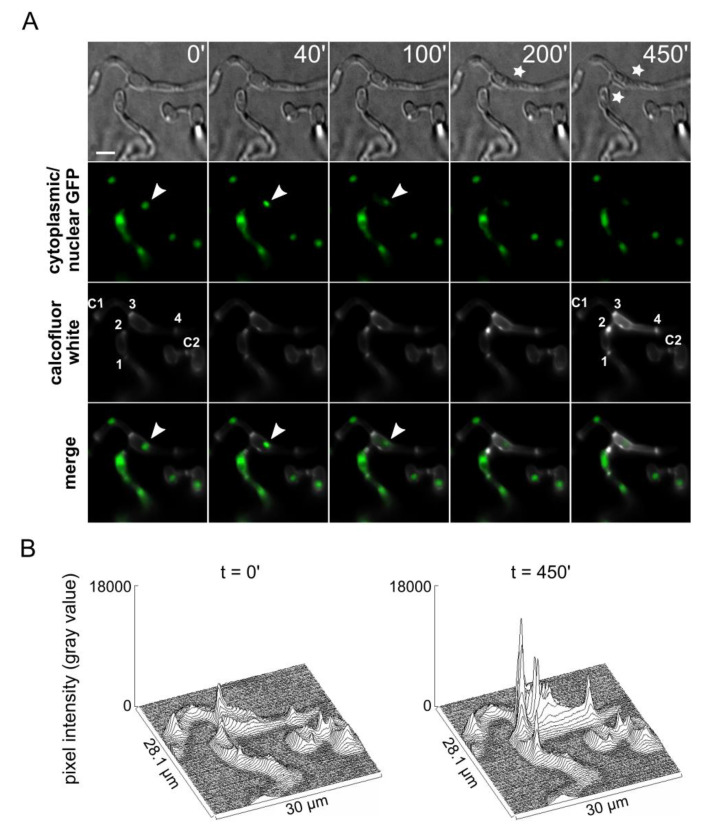
*Vdhex1* is not involved in heterokaryon incompatibility. (**A**) Time-lapse microscopic examination of a fusion event between the incompatible strains BB Δ*hex1* H1-sGFP and Ls.17 sGFP. Typical characteristics of incompatibility-induced cell death, namely, nuclear degradation (arrowheads) and cell shrinkage (asterisks), are observed. Septae that surround the fused cells are indicated by numbers 1–4, and two control septae in neighboring hyphal compartments are indicated by C1–C2 (see also Appendix A). Cell wall staining was performed using calcofluor white M2R. Bar = 10 μm. (**B**) Quantification of blue fluorescence intensity (calcofluor white M2R staining of cell wall chitin) in the region of interest shown in panel A, for two post-fusion time points, before and after the visible effects of the incompatibility reaction.

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
