# Peer review of "Hex1, the Major Component of Woronin Bodies, Is Required for Normal Development, Pathogenicity, and Stress Response in the Plant Pathogenic Fungus Verticillium dahliae"

_jof, 2020, doi:10.3390/jof6040344_

Round 1

Reviewer 1 Report

In this manuscript, the authors investigated a major component of Woronin bodies, Hex1, in the pathogenic fungus Verticillium dahlia and indicated its involvement in normal fungal development, plant pathogenicity and stress response. For this, they identified the Vdhex1 gene, generated a hex1 null-mutant strain and characterized its phenotype in comparison with the wild type strain. Despite that the role of hex1 have been established in other species, the results of this manuscript are interesting due to discrepancies in the importance of the protein in different organisms. I have a couple of concerns with the present work:

  1. While the role of hex 1 in plant pathogenicity appears convincing, in some other aspects the work is rather descriptive and does not seem to add much to the role of hex1. For instance, wouldn’t the authors agree that, in terms of “ROS phenotypes”, other protein mutant strains might show a phenotype similar to the hex1 mutant, without these proteins having a direct role in ROS production or regulation?
  2. It is not clear to me if the rescued strain (hex1-c) was included in all experiments comparing the deletion hex1 mutant with the wild type strain. If not, it should be done.
  3. As compared to the wild type strain, I am not completely convinced of a strong effect of the exposure of the hex1 deletion mutant to different substances (Figures 4-6), because this strong effect is not so apparent in the spot assay images.
  4. How would the authors comment that that the lower sorbitol concentration seem to inhibit more the growth of wild type than that of hex1 (Figure 4B)?
  5. The (lack of) involvement of hex1 in heterokaryon incompatibility apparently contradicts the previous results and knowledge about the role of hex1, namely in septal pore sealing.
  6. The Discussion is rather lengthy and could be shortened. Although I appreciate that it contains a nice review of the subject, it somehow repeats Introduction and Results and seems long for the article.

Reviewer 2 Report

The manuscript is very well written and it includes a good amount of interesting information.

I only have to point out that throughout the Results section there are some overstatements - in my opinion the way some results are described does not completely match what can be seen in the Figures. This is mainly related with the generalization of some facts even when in some situations there is no statistical significance to back them up. I believe it is just a question of re-writing some parts of the text to make them more toned down.

I have also found several inconsistencies in gene and species names format (missing italic).

In the attached pdf file you can find the manuscript with some comments to be addressed.

Round 2

Reviewer 1 Report

Despite that Discussion was not really significantly shortened, I consider that, overall, the authors have coped well with my concerns and performed appropriate alterations.